# Inflammasomes—A Molecular Link for Altered Immunoregulation and Inflammation Mediated Vascular Dysfunction in Preeclampsia

**DOI:** 10.3390/ijms21041406

**Published:** 2020-02-19

**Authors:** Padma Murthi, Anita A. Pinar, Evdokia Dimitriadis, Chrishan S. Samuel

**Affiliations:** 1Cardiovascular Disease Program, Monash Biomedicine Discovery Institute and Department of Pharmacology, Monash University, Victoria 3168, Australia; anita.pinar@monash.edu (A.A.P.); chrishan.samuel@monash.edu (C.S.S.); 2Hudson Institute of Medical Research, Clayton, Victoria 3168, Australia; 3Department of Obstetrics and Gynaecology, University of Melbourne, Parkville, Victoria 3168, Australia; evdokia.dimitriadis@unimelb.edu.au

**Keywords:** preeclampsia, placental insufficiency, inflammasomes

## Abstract

Preeclampsia (PE) is a pregnancy-specific multisystem disorder and is associated with maladaptation of the maternal cardiovascular system and abnormal placentation. One of the important characteristics in the pathophysiology of PE is a dysfunction of the placenta. Placental insufficiency is associated with poor trophoblast uterine invasion and impaired transformation of the uterine spiral arterioles to high capacity and low impedance vessels and/or abnormalities in the development of chorionic villi. Significant progress in identifying potential molecular targets in the pathophysiology of PE is underway. The human placenta is immunologically functional with the trophoblast able to generate specific and diverse innate immune-like responses through their expression of multimeric self-assembling protein complexes, termed inflammasomes. However, the type of response is highly dependent upon the stimuli, the receptor(s) expressed and activated, the downstream signaling pathways involved, and the timing of gestation. Recent findings highlight that inflammasomes can act as a molecular link for several components at the syncytiotrophoblast surface and also in maternal blood thereby directly influencing each other. Thus, the inflammasome molecular platform can promote adverse inflammatory effects when chronically activated. This review highlights current knowledge in placental inflammasome expression and activity in PE-affected pregnancies, and consequently, vascular dysfunction in PE that must be addressed as an interdependent interactive process.

## 1. Introduction

Placental development is complex and is spatio-temporally regulated by an array of factors including placentally- and maternally-derived growth factors, hormones, and cytokines [1]. Uterine spiral arteries are remodeled into highly dilated vessels by the invasion of the extravillous trophoblasts (EVT). Invaded EVTs disrupt the vascular smooth cell layer and replace the endothelium, converting muscular wall arteries into wide bore low-resistance vessels ensuring a local increase in blood supply, which allows for sufficient maternal blood flow into the intervillous space for the nutritional requirements of the fetus [2]. These cellular and physiological homeostatic processes are critical steps in the establishment of a successful pregnancy and occurs in association with an increase in blood volume over the course of pregnancy [3]. Concomitant to the normal physiological adaptation, vascular resistance is also reduced via decreased vascular tone that takes place as the placenta is being formed [4]. An abnormality in the physiological adaptation to normal pregnancy occurs in the common pregnancy-related disorder that endangers the proper development of the embryo/fetus, as well as the health of the mother is preeclampsia (PE). PE is a complex pathophysiological condition where vascular resistance is abnormally increased and contributes to maternal hypertension (Figure 1). Concurrent maladaptation of the maternal cardiovascular system along with abnormal placentation predisposes an individual to PE [5]. There are no cures for PE and removal of the placenta leads to resolution of symptoms of PE, and thus its management mainly relies on delivery, often preterm. Despite the importance of the placenta and the maternal cardiovascular maladaptation in the development of PE, exploring the key molecular pathways is critical for the maintenance of homeostasis in pregnancy and also for identifying novel targets for the prevention and management of PE. This review thereby highlights the importance of inflammasomes as potential molecular links of the key inflammatory pathways underlying the development of PE.

### 1.1. Human Trophoblast Differentiation Establishes the Maternal–Fetal Interface

In a normal pregnancy, the complex architecture at the boundary of the maternal uterus and the fetally-derived placental tissue is governed largely by the differentiation of immunocompetent cytotrophoblasts. Cytotrophoblasts readily undergo differentiation by emigrating from the anchoring villi and joining the cell columns as extravillous cytotrophoblasts (EVTs), serving as conduits to the uterine wall [2,6]. Briefly, the process of invasion of the EVTs in the maternal decidua and inner third of the myometrium occurs via two spatial routes. Firstly, migration and invasion through the decidual layer gives rise to interstitial EVTs that interact with specialized populations of maternal immune cells present within the decidua. Secondly, the EVTs that move up and target the lumen of the spiral arteries become endovascular EVT, adopt an endothelial-like phenotype, and intercalate within the smooth muscle cells of the tunica media. The remodeling of the spiral arterial wall involves loss of normal musculoelastic structure and replacement of the vascular media with fibrinoid material. This results in converting the originally low capacitance/high resistance uterine arteries into high capacitance low resistance channels that perfuse the surface of the placenta, which is enveloped by multinucleated transport epithelium, namely syncytiotrophoblast (STB) [1]. The invading EVTs exhibit a major transformation at both the cellular and molecular levels. Cytotrophoblasts attached to the chorionic villi initially express adhesion molecules characteristic of epithelial cells such as integrins α6/β1, αv/β5. As cytotrophoblasts enter the cell columns and adopt the invasive pathway, the expression of epithelial cell-like adhesion molecules decrease, whilst the expression of endothelial cell adhesion markers such as integrins α1/β1, αv/β3, and VE-cadherin is upregulated to promote vascular mimicry [2,7].

### 1.2. Preeclampsia

Preeclampsia (PE) is a serious human pregnancy-specific disorder, although clinically defined as a new onset of hypertension and proteinuria, in the absence of proteinuria, PE may be diagnosed with any of the following features: Thrombocytopenia (platelet count less than 100,000 × 109/L); impaired liver function as indicated by abnormally elevated blood concentrations of liver enzymes (to twice the upper limit of normal concentration); severe persistent right upper quadrant or epigastric pain and not accounted for by alternative diagnoses; renal insufficiency (serum creatinine concentration greater than 1.1 mg/dL or a doubling of the serum creatinine concentration in the absence of other renal disease); pulmonary edema; or new-onset headache unresponsive to acetaminophen and not accounted for by alternative diagnoses or visual disturbances. It should also be noted that these criteria must be met after the 20th week of gestation, otherwise this would be classed as chronic hypertension.

PE affects 2%–8% of pregnancies, is associated with 12% of infants with fetal growth restriction (FGR) and approximately 20% of preterm deliveries [8]. PE remains lacking reliable means of diagnosis and prediction, with no effective therapy or pharmacological agents available to treat the disease. The only solution is the removal of the placenta by early delivery and often preterm delivery. As depicted in Figure 1, if left untreated, PE can lead to life threatening systemic vascular dysfunction and may progress to eclampsia with complications of the HELLP syndrome (elevated liver enzymes, haemolysis, and low platelets), placental abruption, acute renal failure, and pulmonary edema [9]. Although maternal symptoms appear to be largely resolved with the delivery of the placenta and the fetus, accumulating evidence suggests that PE is associated with long-term maternal cardiovascular and other complications such as renal diseases [10,11,12,13,14]. 

#### 1.2.1. Placental Ischemia 

Poor placentation resulting from the insufficient EVT invasion and reduced uterine blood flow leading to placental ischemia have been proposed to be one of the leading causes of PE. Specifically, placental ischemia is associated with an imbalance of immune function and chronic inflammation similar to autoimmune diseases [15,16,17]. This immune imbalance creates a state of chronically uncontrolled inflammation due to an increase in the production of pro-inflammatory factors and the abnormal activation of immune cells and production of cytokines, whilst decreasing the abundance of regulatory immune cells and cytokines [18,19]. These alterations lead to progressive pathophysiological changes including an enhanced production of reactive oxygen species [20], increased oxidative and endoplasmic reticulum stress, endothelin-1 expression, production of potent pro-inflammatory mediators [21], anti-angiogenic factors at the implantation site, and the production of autoantibodies such as angiotensin-II type I receptor (AT1-AA) [22], thus culminating in the development of hypertension during pregnancy. Therefore, insight into how different components of the innate immune system and inflammatory cytokines are regulated and contribute to the progression of development of PE will be useful in developing targeted therapies that are necessary to improve pregnancy outcomes. 

#### 1.2.2. The Immunology of Pregnancy

Human pregnancy is an immunological paradox [23]. Maintenance of pregnancy relies on finely tuned immune adaptations at the maternal-fetal interface where the two distinct genomes of the mother and the fetus commingle to maintain tolerance to the fetal allograft while preserving innate and adaptive immune mechanisms for protection against microbial challenges. Studies by Co et al. [23] have reported that trophoblast interaction with the decidual natural killer cells (dNK) is crucial for the maternal-fetal immune tolerance early in pregnancy. Combinations of signals and responses originating from the maternal and fetal-placental immune systems are critical for a successful placentation, as well as for pregnancy outcome [24]. The signals that originate from the placenta has the ability to sense the infectious and non-infectious triggers and generate innate, immune-like responses [25]. The placenta not only uses several mechanisms to regulate immune tolerance and adaptation, but it may also modulate the way the maternal immune system adapts in the presence of potential dangerous signals [25,26]. Immunologic miscommunications that have origins at the placenta or in the mother may contribute to disruption to the regulatory and protective mechanisms leading to pregnancy complications including PE. 

#### 1.2.3. Inflammation in the Development of PE

A generalized systemic inflammatory response is common to all pregnancies, as highlighted by Redman et al. [27] which proposed that PE is intrinsically similar to normal pregnancy and is characterized by the extreme end of a continuous spectrum of inflammatory responses. In PE, a deviation in the physiological immunoregulatory adaption to pregnancy has been described for promoting inhibitory reactivity to the fetus. Furthermore, in PE-affected pregnancies, an increase in immune cells has been demonstrated in response to activation of the innate immune system and inflammation in the maternal circulation and uteroplacental unit [28]. These events in turn contribute to the shallow invasion of EVT in the uterine wall and insufficient spiral arteriole remodeling leading to placental ischemia (Figure 1). 

As a consequence of placental ischemia, oxidative stress is augmented with an excessive release of placental factors, such as STB knots/debris, soluble fms-like tyrosine kinase-1 (sFlt-1), the soluble receptor for vascular endothelial growth factor (VEGF) into the maternal circulation, which collectively contribute to the development of hypertension [29,30]. These angiogenic factors are also critical inflammatory mediators which contribute to maternal inflammation associated with PE [31]. Villous cytotrophoblasts mediate inflammation via the secretion of inflammatory cytokines including interleukins (ILs)-1β, -2, -4, -6, -8, -10, -12, and -18, transforming growth factor (TGF)-β1, IFN-γ-inducible protein 10/IP-10, tumor necrosis factor (TNF)-α, interferon (IFN)-γ, monocyte chemotactic protein (MCP)1, intercellular adhesion molecule (ICAM)-1, and vascular cell adhesion molecule (VCAM)-1 [6,28,32,33,34], which contribute to the development of PE. This role is particularly important for cytotrophoblasts, which fuses to form STB, and is the site for maternal–fetal interactions. Several cytokines including IL-1β and IL-18 have also been correlated with maternal endothelial dysfunction [34]. These inflammatory cytokines can activate several downstream pathways both directly or indirectly to contribute to the clinical manifestations and progression of PE. Several triggers including elevated maternal serum concentrations of cholesterol and uric acid in PE have been shown to contribute to a heightened inflammatory response through STB [35]. Depending on the stimulant or physiological change(s) that occur during pregnancy, placental cells may mount either a regulated protective response that maintains and promotes a healthy pregnancy, or alternatively, promotes a damaging response that adversely impacts the outcome of pregnancy, as observed in PE. One such molecular mechanism by which inflammatory responses are regulated in the human placenta is via the molecular inflammasome platform [36], which are known producers of IL-1β and IL-18. A greater understanding of the precise molecular pathways governed by inflammasomes in the placental sensing mechanisms is therefore critical for drug discovery and therapeutic targeting.

As a consequence of placental ischemia, oxidative stress is augmented with an excessive release of placental factors, such as STB knots/debris, soluble fms-like tyrosine kinase-1 (sFlt-1), the soluble receptor for vascular endothelial growth factor (VEGF) into the maternal circulation, which collectively contribute to the development of hypertension [29,30]. These angiogenic factors are also critical inflammatory mediators which contribute to maternal inflammation associated with PE [31]. Villous cytotrophoblasts mediate inflammation via the secretion of inflammatory cytokines including interleukins (ILs)-1β, -2, -4, -6, -8, -10, -12, and -18, transforming growth factor (TGF)-β1, IFN-γ-inducible protein 10/IP-10, tumor necrosis factor (TNF)-α, interferon (IFN)-γ, monocyte chemotactic protein (MCP)1, intercellular adhesion molecule (ICAM)-1, and vascular cell adhesion molecule (VCAM)-1 [6,28,32,33,34], which contribute to the development of PE. This role is particularly important for cytotrophoblasts, which fuses to form STB, and is the site for maternal–fetal interactions. Several cytokines including IL-1β and IL-18 have also been correlated with maternal endothelial dysfunction [34]. These inflammatory cytokines can activate several downstream pathways both directly or indirectly to contribute to the clinical manifestations and progression of PE. Several triggers including elevated maternal serum concentrations of cholesterol and uric acid in PE have been shown to contribute to a heightened inflammatory response through STB [35]. Depending on the stimulant or physiological change(s) that occur during pregnancy, placental cells may mount either a regulated protective response that maintains and promotes a healthy pregnancy, or alternatively, promotes a damaging response that adversely impacts the outcome of pregnancy, as observed in PE. One such molecular mechanism by which inflammatory responses are regulated in the human placenta is via the molecular inflammasome platform [36], which are known producers of IL-1β and IL-18. A greater understanding of the precise molecular pathways governed by inflammasomes in the placental sensing mechanisms is therefore critical for drug discovery and therapeutic targeting.

## 2. Inflammasomes

The ability to activate immune cells depends on the presence of molecular multiprotein inflammasome complexes [37,38]. Inflammasomes are high molecular-weight multimeric self-assembling protein complexes of the innate immune system, which are not only important for initiating the inflammatory response and release of IL-1β and IL-18 (Figure 2), but also in regulating cellular apoptosis [38]. As shown in the schematic diagram, signaling for inflammasome activation occurs in two stages. Priming in stage 1 is followed by activation of inflammasome complex by interaction with several components. The inflammasome complex contains a sensor molecule, an adaptor protein and the pro-inflammatory caspase-1 [37,39]. Inflammasomes act as a finely tuned alarm, triggering and amplifying systems in response to cellular stresses and/or infections. Placental trophoblasts, endothelial cells and macrophages can sense and respond to a variety of infectious agents by the presence of the pattern recognition receptors (PRRs). The PRRs have the ability to sense pathogen-associated molecular patterns (PAMPs) that are expressed by microbes, as well as noninfectious (sterile inflammation) host-derived damage associated molecular patterns (DAMPs) including reactive oxygen species, uric acid, cholesterol, microparticles, and exosomes (namely alarmins) [40,41] (Figure 2). When activated, the inflammasome machinery has the potential to promote maturation and release of pro-inflammatory cytokines including the release of danger signals, as well as pyroptosis, a rapid, pro-inflammatory form of cell death [42].

### 2.1. Pattern Recognition Receptors (PRRs)

The potential for a cell to promote inflammation is evident by its expression of a repertoire of PRRs. Two major families of PRRs are the Toll-like receptors (TLRs) and the nucleotide-binding domain leucine-rich repeat containing receptors [37], also known as Nod-like receptors (NLRs) (Figure 3). TLRs are transmembrane receptors, which allow for the sensing of PAMPs or DAMPs either at the cell surface or within endosomal compartments. There are 10 human TLRs that have been identified, each with distinct specificities to activate downstream cascade of events in response to Gram-positive/Gram-negative bacterial infection or viral RNA [43,44]. NLRs are cytoplasmic-based PRRs, which provides an intracellular recognition system for sensing PAMPs or DAMPs. Endogenous danger signals initiate and maintain inflammatory responses through activation of NLRs (Figure 3). 

Multiple NLRs and NLR-dependent inflammasomes have been identified including pyrin-domain containing the initiator proteins including NLRP-1, NLRP3, NLR family caspase activation, and recruitment domain (CARD) domain-containing protein-4 (NLRC4) and the adaptor protein called ASC (apoptosis-associated speck-like protein containing a CARD) [45]. Additionally, NLR-independent inflammasomes that are driven by sensor molecules such as absent in melanoma-2 (AIMS2) and pyrin have also been described [46]. In the absence of TLRs or reduced signaling of TLRs, NLRs synergize with TLRs for a greater response or provide a compensatory system [46]. Several of the sensor molecules of the inflammasomes such as NLRP1, NLRP3, NLRP7, NLRC4, AIM2, and pyrin have been well characterized for their specific ligands, mechanisms of action, and roles in disease pathogenesis (reviewed in Strowig et al. [47]). However, inflammasomes such as NLRP6, NLRP12, retinoic-acid inducible gene-1 (RIG-I), and interferon-γ inducible protein-16 (INFI16) are yet to be fully characterized in health and disease. 

### 2.2. Inflammasome Components in the Gestational Tissues during Normal Pregnancy

An inflammatory environment is mandatory in order to ensure an adequate reconstruction of the uterine epithelium, elimination of cellular debris, and tissue remodeling during implantation, placentation, maintenance of pregnancy throughout gestation and in parturition [48,49,50]. Inflammasome components have been identified in both maternal and fetal compartments throughout gestation. The mRNA transcripts, as well as the protein expression of NLRP1-4; the adaptor protein ASC (or PYCARD) and the caspases (1 and 4) have all been detected in the human placental trophoblasts, myometrium, and in the amniotic membranes throughout gestation [51,52,53,54,55,56,57]. Hyperactivity of inflammasomes has also been reported in the choriodecidua, myometrium, and cervix during term parturition (reviewed in [36]). Specifically, several studies indicate that the activation of NLRP3 inflammasome leads to the pyroptosis as part of the sterile inflammatory milieu during physiological labour in term pregnancies [58]. Emerging studies describing inflammasome expression and activation in physiological inflammation associated with uncomplicated human pregnancies provide important knowledge on their role in the pathophysiology of pregnancy complications such as PE [35] and FGR [59]. 

### 2.3. Activation of Inflammasomes in Preeclampsia-Affected Pregnancies

Activation of inflammasomes are an essential element of the innate immune system and disturbances in these processes have been implicated in various inflammatory diseases including placental inflammation associated with PE [35], FGR [59], and gestational diabetes mellitus [60]. Inflammasome hyperactivity has been reported in placental tissues from pregnant women with PE [35,41,61]. Mulla et al. [62] reported that inflammasome activation in STB could be a possible mechanism of induction of inflammation at the maternal–fetal interface that causes adverse pregnancy outcomes, including PE. To further support this, elevated levels of TLR2, TLR4, NLRP3, and IL-1β have been reported in the neutrophils of women with PE, when compared to normal pregnant women [44]. Furthermore, Pontillo et al. [52] reported enhanced gene transcripts for NLRP1, NLRP3, NLRC4, ASC, caspase-1, and IL-1β following stimulation with lipopolysaccharide (LPS) in human first-trimester cytotrophoblasts, decidual stromal cells, and endothelial cells in vitro. Anti-angiogenic factor (sFlt-1) in the syncytial knots and TNFα release from the STB were also implicated in inducing higher inflammasome activation in the placental tissues from PE pregnancies [35,41]. Further to this, an association between higher levels of TNFα and NLRP3 activation in peripheral blood monocytes from PE pregnancies demonstrated a direct involvement of TNFα in inflammasome activation [63]. 

Recent studies by Nunes et al. reported that alterations in STB functions as a consequence of the imbalance between pro- and antioxidant properties may also cause cellular stress and injury to activate inflammasomes [64]. Although the molecular mechanism involved in placental inflammasome activation in PE is largely unknown, it is possible that the inappropriate inflammatory response observed in PE may have its origin in the placenta. Potential mechanisms may include shedding of STB-derived micro or nanovesicles (that can act as DAMPs) into the maternal circulation, which are known to exert pro-inflammatory, procoagulant, and anti-endothelial activity in vitro [40,65]. Recent studies [41,65] also demonstrated that oxidative stress induced increase in the release of high-mobility group box 1 protein (HMGB1) from STB that may contribute to the pathogenesis of PE. Furthermore, Ivernsen (2013) [66] reported that the inflammatory molecules such as heat shock protein 70 (Hsp70), HMGB1, Galectin 3, and Synctin 1 carried by microvesicles in PE pregnancies may also act as DAMPS in the placenta and in the peripheral blood mononuclear cells (PBMC) in patients with PE. Thus, as illustrated in Figure 3, placental milieu resultant from the STB activation by inflammatory cytokines/or release of microparticles from injured or necrotic cells or complement primed and endogenous uric acid accumulation can activate the inflammasome machinery [35,61,62,63,67]. 

Although PE has been considered the disease of the primigravid for many years, a robust evidence for this important observation has never been reported. Several studies have also reported that there is a role for the immune system during pregnancy reacting against and/or tolerating the paternal antigens of the conceptus [68,69,70,71,72,73,74]. It was suggested that that increased exposure to the father’s semen assists this immunological tolerance [68]. In addition to these benefits, although semen is not sterile, microbial tolerance mechanisms may exist [75]. Recent reports [68,75] have shown evidence that semen may be responsible for inoculating the developing conceptus, including the placenta with microbes, not all of which are infectious. It was suggested that when they are infectious, it may cause PE [68]. Furthermore, a variety of epidemiological and other evidence is entirely consistent with this, not least correlations between semen infection and PE [71,76]. Overall, these studies strongly suggest a significant paternal role in the development of PE through microbial infection of the mother via insemination. 

Taken together, the above studies suggest that inflammasome activation may play a central role in the placental inflammatory processes that are associated with the pathophysiology of pregnancy complications including PE. However, further studies are required to investigate whether the inhibition of inflammasomes can be considered as a potential therapeutic strategy to prevent placental inflammation-induced development of PE.

## 3. Inflammasome-Mediated Downstream Molecular Pathways in Long-Term Vascular Functions 

There is evidence that women who have PE and pregnancy-induced hypertension or who deliver a preterm baby have an increased risk of developing cardiovascular disease later in life [10,77]. Several systematic reviews and meta-analyses have determined that after a diagnosis of PE the relative risks for developing hypertension, cardiometabolic disorders are significantly increased in both mother and in child [78,79,80]. Follow-up studies of children who were born prematurely show evidence of an increased risk of high blood pressure and insulin resistance in their adulthood [81]. Thus, these reports suggest that the pregnancies associated with PE, pregnancy hypertension; as well as pregnancies associated with preterm delivery show evidence of significant changes in vascular function (detailed below) at the time of the pregnancy. These changes greatly impact the cardiovascular health of both the mother and child later in life. The cascade of events downstream of inflammasomes may play a critical role in changes associated with vascular functions in hypertensive disorders [82] including PE; however, the molecular mechanisms by which inflammasomes promote pathogenesis of PE is yet to be investigated. In the following section, we provide possible mechanisms of inflammasome-mediated alterations observed in hypertensive disorders, which may provide a foundation for developing improved management and treatment strategies to reduce pregnancy specific burden of vascular dysfunction associated with PE.

### 3.1. Activation of Inflammasomes in Hypertensive Disorders

Hypertension, defined as individuals presenting with blood pressure greater than 140/90 mmHg [83], represents a worldwide-spread cardiovascular abnormality and is a major cause of subsequent end-organ damage observed in affected patients. Numerous studies have detailed the pathogenesis of hypertensive disorders and reported that the molecular inflammasome platform represents a central pathogenic mechanism in initiating and promoting organ damage attributed to hypertension. Eight-week-old male Dahl salt-sensitive rats fed with a high-salt diet (8% NaCl) for six weeks were found to have higher levels of NLRP3 and IL-1β in the hypothalamic paraventricular nucleus when compared to rats fed a normal diet (0.3% NaCl) [84]. Similarly, bilateral hypothalamic paraventricular nucleus injection of an IL-1β inhibitor, gevokizumab (1 µL of 10 µg), reduced the mean arterial pressure, heart rate, and levels of plasma norepinephrine, as well as, attenuated the levels of oxidative stress (NOX-2 and NOX-4) and restored the levels of NLRP3, IL-1β, and IL-10 [84]. Additionally, via inhibiting NLRP3-induced inflammation and idiopathic pulmonary fibrosis, the clinically used TGF- β blocker, pirfenidone protected against thoracic aortic constriction (TAC)-induced hypertension and left ventricular hypertrophy, collectively contributing to myocardial fibrosis, via blocking NLRP3-mediated inflammation and fibrosis [85]. 

Furthermore, the ASC adaptor protein of the NLRP3 inflammasome, was shown to be critical in hypoxia-induced pulmonary hypertension and right ventricular remodeling which was associated with increased protein levels of caspase-1, IL-18, and IL-1β [86]. Moreover, Asc*^−/−^* mice demonstrated reduced collagen deposition and muscularization around arteries [86]. Collectively, the findings from this study indicate that hypoxia promotion of right ventricular pressure and remodeling were attenuated in mice lacking Asc, but not in mice lacking Nlrp3, indicating that the inflammasome molecular platform plays a critical role in the pathogenesis of pulmonary hypertension [86]. Another study reported that 1K/DOCA/salt-induced hypertensive mice demonstrated increased expression of renal Nlrp3, Asc, and pro-caspase-1, as well as IL-1β and IL-18 mRNA [87]. Additionally, Asc*^−/−^* mice in the same model were protected from an increase in the renal inflammatory profile (IL-6, IL-17A, CCL2, ICAM-1, and VCAM-1) and accumulation of macrophages and collagen [87]. These studies suggested that the cascade of events downstream of inflammasomes play a critical role in disease progression; their mechanism of actions include both a central nervous and a peripheral modulation of the inflammatory pathways.

### 3.2. Inflammasomes: A Potential Molecular Link for Long-Term Vascular Dysfunction and End-Organ Failure in Preeclampsia 

The villous stroma of the placenta provides the microenvironment for placental vascular development where immune cells reside and serve as a barrier to induce inflammatory (inflammasome)-mediated responses [88]. PE involves the excessive activation of inflammatory immune cells [63], including monocytes, fibroblasts, and granulocytes and their exacerbated production of pro-inflammatory cytokines, IL-1β, IL-6, and IL-8 [89,90], and reduced production of regulatory cytokines such as IL-10 and TGF-α [91]. In this setting, TGF-β-promoted extracellular matrix (ECM) proteins, such as collagens, laminins, and fibronectin, play a key modulatory role in tissue remodeling [88,92,93]. Placental fibroblasts modulate the expression of ECM proteins (collagens I and IV, fibronectin, and fibrillin I) more prominently in the first trimester and term tissue [88]. Placental ischemia primes aberrant vascular and uteroplacental remodeling via the release of pro-inflammatory factors cytokines such as TNF-α in the maternal circulation [94,95,96,97]. Li et al. quantified the levels and distribution of MMPs measured in the aorta, uterus, and placenta of normal versus pregnant rats with reduced uterine perfusion pressure (RUPP) [94]. Gelatin zymography showed marked levels of uterine MMP-2 and MMP-9, whereas casein zymography demonstrated upregulated MMP-1 and MMP-7 in the aorta, uterus, and placenta of pregnant rats with reduced uterine perfusion pressure, compared with that from normal pregnant rats. Supplementary organ culture work in the same study demonstrated that TNF-α stimulation upregulated the levels of MMP-1 and MMP-7 in the aorta, uterus, and placenta of normal pregnant rats, whereas a TNF-α inhibitor antagonized the increased tissue MMP levels in rats with RUPP [94]. 

Collectively, these findings suggest that placenta ischemia, via TNF-α mediated signal transduction and potentially through priming of the inflammasome platform, could lead to inadequate uteroplacental and aberrant vascular remodeling in pregnancies associated with hypertension and PE. Targeting MMP-1 and MMP-7, and/or the TNF receptor upstream of that, may also present a novel avenue in the therapeutic modulation of inflammasome priming that promotes hypertension and PE [94]. As previously discussed, women with PE also demonstrate an elevated hyperuricemia profile associated with proteinuria, suggesting that increased levels of uric acid promote the disease severity and pathogenesis associated with PE, via inducers of the NLRP3 inflammasome [90]. Uric acid is known to promote inflammation and endothelial dysfunction [98] and its crystals, monosodium urate (MSU) promote the release of IL-1β via activation of the NLRP3 inflammasome (Figure 2) [35,38,63,87]. Monocytes from PE women were activated and hence released higher levels of TNF-α, superoxide anion (O_2_^−^), and H_2_O_2_ compared to monocytes derived from normotensive pregnant women [99]. These findings indicated that monocytes from the maternal peripheral blood are a key source of reactive oxygen species, free radicals, and pro-inflammatory cytokines. Collectively, these studies suggest that the production of IL-1β, via activation of the inflammasome cascade, is key to driving the pathogenesis of PE. To date, research is trying to design inflammasome antagonists or equivalent inhibition strategies.

## 4. Therapeutic Targeting for the Components of Inflammasomes

Due to the wide range of hypertension-driven inflammatory diseases, a number of targeted therapies have been investigated for antagonizing the effects of the inflammatory (inflammasome)-pathway. Pharmacological blockade of human cord blood leukocytes demonstrated that ATP released as a result of tissue injury can further promote the secretion of IL-1β from laboring and nonlaboring women [100], indicating that inhibition of the P2X7 receptor may protect from inflammation induced vascular injury during pregnancy. A separate study evaluated pharmacological blockade using a P2X7 receptor inhibitor and pannexin-1 blocker carbenoxolone, was shown to attenuate the LPS-induced increase in the levels of secreted IL-1β [101]. Silibinin (SB) is a flavonoid complex medicinal herb with anti-inflammatory, hepatoprotective, antioxidant, and antifibrotic properties. The antioxidant and anti-inflammatory properties of SB were assessed via dose-dependent inhibition of H_2_O_2_ release, production of TNF-α, IL-10, TGF-β, and prostaglandin E2 (PGE2) following LPS stimulation of peripheral blood monocytes from healthy individuals. Monocytes were treated with SB to determine whether SB can modulate the NLRP1 and NLRP3 inflammasomes, as well as influence upstream TLR-4/NF-κB activation [99]. Administration of SB to MSU-stimulated monocytes reduced the degree of NLRP1 and NLRP3 inflammasome activation, as well as TLR-4/NF-kB activation [99]. Furthermore, administration of SB to pregnant rats in an experimental model of PE, induced by nitric oxide synthase inhibition (with N-omega-nitro-l-arginine methyl (l-NAME)) protected reproductive outcomes, normalized blood pressure, reduced proteinuria, and also serum levels of pro-inflammatory cytokines [102]. Additional studies have examined targeting of the NLRP3 sensor itself, via administration of the small and highly selective NLRP3 inflammasome inhibitor, MCC950, a diarylsulfonylurea-containing compound [103]. The mechanism of action of MCC950 is via its ability to inhibit NLRP3-induced ASC oligomerization to subsequently block the secretion of IL-1β [103], where MCC950 was recently shown to attenuate the high deoxycorticosterone (DOCA)-induced hypertensive effects [87,104]. Krishnan et al. [87] demonstrated an increased DOCA/salt-induced renal inflammatory profile (IL-6, IL-17A, CCL2, ICAM-1, and VCAM-1), fibrosis (assessed via extent of renal collagen accumulation), via an inflammasome/IL-1β-dependent mechanism. 

As a follow-up study, Krishnan et al. [104], showed that pharmacological inhibition of the NLRP3 inflammasome, with MCC950, significantly lowered the 1K/DOCA/salt-induced increase in blood pressure, renal expression of inflammasome markers (NLRP3, ASC, pro-caspase-1, pro-IL-1β and pro-IL-18), and markers associated with renal inflammation and injury (IL-17A, TNF-α, osteopontin, ICAM-1, VCAM-1, CCL2, and vimentin). A MCC950-induced reduction in these 1K/DOCA/salt-induced measures was accompanied by an additional marked attenuation in the levels of renal interstitial collagen and renal albuminuria (by up to 25%) in C57Bl/6 mice [104]. Similarly, by blocking the ability of the NLRP3 inflammasome components to assemble and oligomerize, as well as inhibiting K^+^ efflux, β-hydroxybutyrate (BHB) was shown to reduce the production of both IL-1β and IL-18 [105]. A separate study also demonstrated that treatment with EMD638683, a specific glucocorticoid-inducible kinase (SGK1) inhibitor, significantly reduced hypertension induced cardiac damage [106]. Taken together, these studies provide proof-of-concept that pharmacological inhibition of upstream, as well as downstream targets of the NLRP3 inflammasome signaling cascade; and the inflammasome platform, present a viable anti-hypertensive strategy in attenuating the pathogenesis of PE with an underpinning inflammatory component. 

## 5. Conclusions

There is consistent evidence for activation of inflammasomes in physiological inflammatory processes during pregnancy. Inflammasomes are implicated in both normal physiological and in the pathophysiological processes that occur in response to inflammatory milieu throughout gestation. This suggests that the placenta is immunologically functional and able to generate specific and diverse innate immune-like responses through the expression of specific components of inflammasomes. However, the type of response is highly dependent upon the timing of gestation, as well as the type of stimuli, the expression and activation of status of the specific type of receptors, and the downstream signaling pathways that are altered in response to the stimuli. Furthermore, it is in the pathological inflammatory processes as seen in PE, that premature activation of inflammasomes as a consequence of placental ischemia (Figure 4) could contribute to increased fibrosis resulting in end organ damage in the heart, liver, and kidneys. Therefore, research findings in these areas will be critical for developing targeted therapies for the prevention of poor pregnancy outcomes of PE affected pregnancies. 

## Figures and Tables

**Figure 1 ijms-21-01406-f001:**
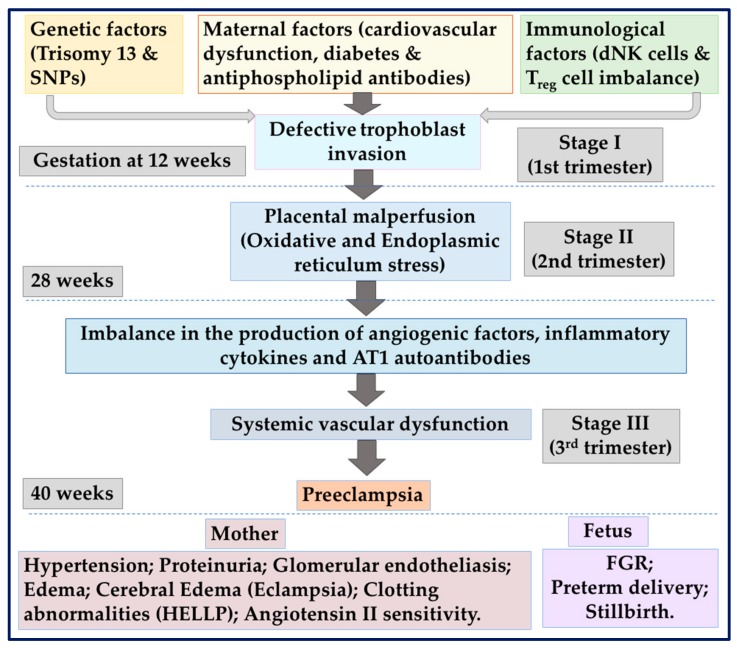
Depicts the complex gestation and stage specific pathophysiological processes associated with preeclampsia. Genetic factors, maternal factors, and immunological factors may cause placental dysfunction (stage I), which in turn leads to the release of anti-angiogenic and other inflammatory mediators that induce preeclampsia (PE) (stage II). AT1: Angiotensin II type I receptor; dNK: Decidual natural killer; HELLP: Haemolysis, elevated liver enzymes and low platelet count; SNP: Single-nucleotide polymorphism; Treg: Regulatory T-cell.

**Figure 2 ijms-21-01406-f002:**
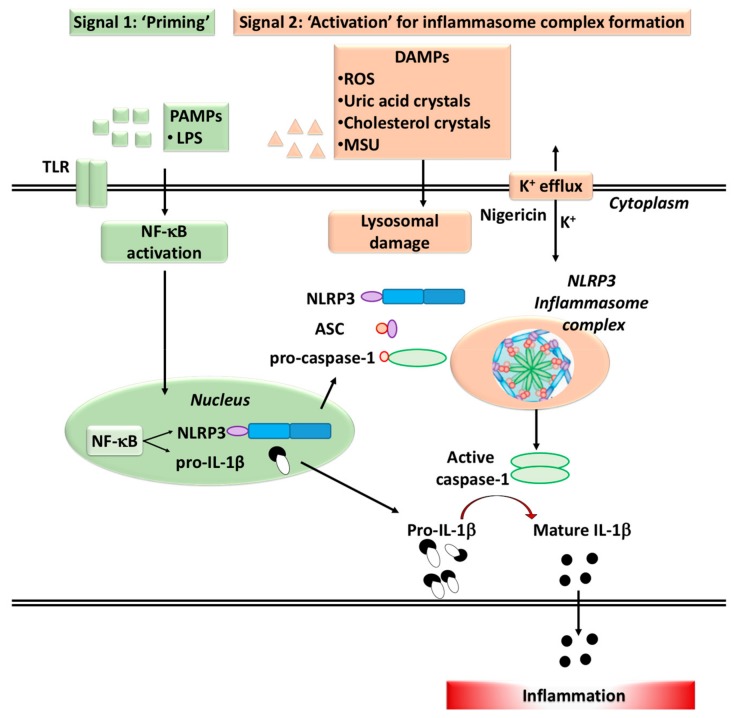
Activation and formation of inflammasome complex. Two-step signaling mechanisms are involved in the formation inflammasome complexes: The first “priming” signal initiated by infectious agents such as LPS, enhances the expression of inflammasome components and target proteins via activation of transcription factor NF-κB. The second “activation” signal promotes the assembly of inflammasome components. The second signal also involves three major mechanisms, including lysosomal damage, and the potassium efflux. Nigericin is a microbial toxin that alters potassium efflux. Abbreviations: LPS: Lipopolysaccharide; TLR: Toll-like receptor; PAMPs: Pathogen-associated molecular patterns; DAMPs: Danger-associated molecular patterns; ROS: Reactive oxygen species.

**Figure 3 ijms-21-01406-f003:**
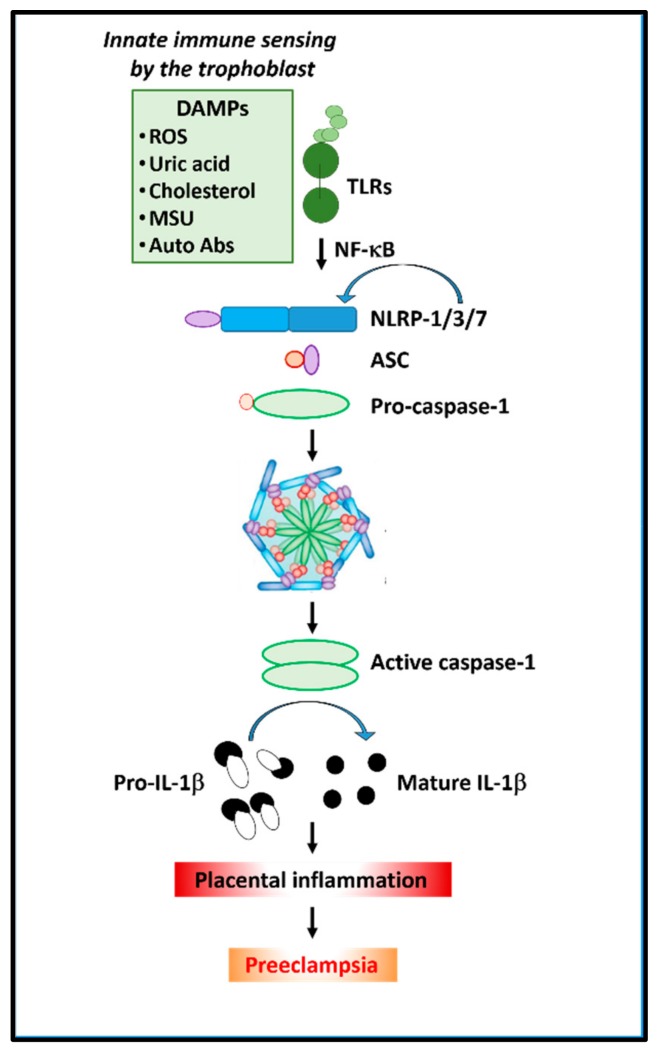
Depicts the role of inflammasomes in placental inflammation in preeclampsia. Cholesterol or uric acid crystals (alarmins) and extracellular vesicles/microparticles can trigger NLRP (1/3/7) inflammasomes in the placenta, which releases active caspase-1 and mature IL-1β and increases inflammation.

**Figure 4 ijms-21-01406-f004:**
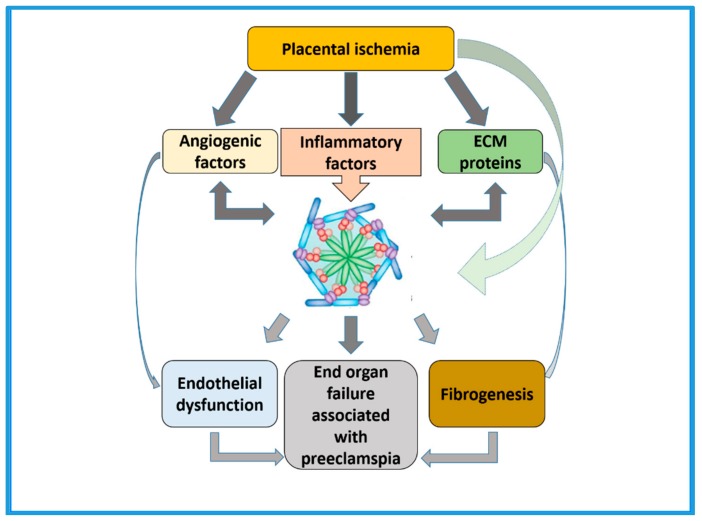
Summarizes the central role of inflammasomes in the pathogenesis of preeclampsia. Premature activation of inflammasomes following placental ischemia in preeclampsia affected pregnancies may lead to changes in vascular structure and function and enhanced fibrosis contributing to end stage organ failure.

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
