# Peer review of "Inflammasomes—A Molecular Link for Altered Immunoregulation and Inflammation Mediated Vascular Dysfunction in Preeclampsia"

_ijms, 2020, doi:10.3390/ijms21041406_

Round 1
Reviewer 1 Report
Figure 1 would require richer description. Authors should point out most significant processes shown in the diagram. Picture has to be standalone, comprehensive even without the article itself Row 106 - authors should explain statement that generalised inflammation is common to all pregnancies. Does it mean that pregnancy is an illness or it is disputed there that a kind ob subtle below-threshold reactions resembling inflammatory response are present? Chapter 2 - a picture of an inflammasome would be very illustrative thus increasing the value of this article Row 215 "pro-and antioxidant" - insert space between "pro-" and "and" Rows 259 - 262 require a quotation (definition of hypertension) Formal aspect would require better uses of paragraphs, as these blocks of text are hardly readable decreasing comprehensibility of the text.Author Response
Please see attachment

Reviewer 2 Report
This is a valuable overview on a long neglected topic in periconceptional and perinatal physiology: the molecular effects of immunologic communication between mother and conceptus and their role in the development of the so called “placental syndromes”. The paper is well written, the presentation is structured, the content up to date.
My major comments are:
On several occasions thoughout the manuscript, it is stated that a dysfunction of the placenta is the cause op preeclampsia. Currently, a vivid debate is ongoing whether this is actually true. More and more arguments are being published now on pre-conceptional maternal cardiovascular dysfunctions, predisposing to an abnormal placentation process. As such, at least in a proportion of cases, the dysfunction of the placenta is a consequence rather than a cause of maternal cardiovascular disorders during pregnancy. The statement “placenta is the cause of preeclampsia” should be tempered throughout abstract and body of the text.
Similarly, on several occasions throughout the manuscript and in the Figures, hypoxia is disscussed as a cause of oxydative stress of trophoblast cells. It should be emphasised that this hypoxia has never been documented in experimental research (Huppertz B, Weiss G & Moser G ( 2014). Trophoblast invasion and oxygenation of the placenta: measurements versus presumptions. J Reprod Immunol 101–102, 74–79.) and as such is also questioned in many papers on periconceptional physiology. The statement on placental hypoxia should be tempered as well in both text and figures.
In line with the above comments, the manuscript presents the communication between placenta and mother as a one way traffic. However, as in any system of communication, errors may occur both at the side of the sender or the receiver. So, immunologic miscommunications may have their origin at the placenta but also in the mother. The latter is currently missing in this review.
For ages, preeclampsia has been considered the disease of the primigravid. However, a clear explanation for this important observation has never been reported. I think this paper contains all contents to do so, as it is very likely that there is a role for the immune system reacting against and/or tolerating the paternal antigens of the conceptus. The authors could elaborate more extensively on this in an extra paragraph?
Reviewer 3 Report
Overall the manuscript by Murthi and colleagues is a good overview of the role of inflammasomes in preeclampsia and other hypertensive disorders. The manuscript is well organized and written, and provides a strong overview of the topic. However there are some minor issues to address.
1) The introduction paragraph, specifically lines 45-47, contain a few grammatical errors such as missing words.
2) Lines 78-79 define preeclampsia (PE) as "new-onset hypertension and proteinuria." This definition is incomplete and somewhat outdated. The American College of Obstetricians and Gynecologists updated their guidelines initially in 2013, and subsequently refined in 2019 to expand the clinical definition of PE. In the absence of proteinuria, PE may be diagnosed with any of the following features: thrombocytopenia (platelet count less than 100,000 × 109/L); impaired liver function as indicated by abnormally elevated blood concentrations of liver enzymes (to twice the upper limit of normal concentration); severe persistent right upper quadrant or epigastric pain and not accounted for by alternative diagnoses; renal insufficiency (serum creatinine concentration greater than 1.1 mg/dL or a doubling of the serum creatinine concentration in the absence of other renal disease); pulmonary edema; or new-onset headache unresponsive to acetaminophen and not accounted for by alternative diagnoses or visual disturbances. It should also be noted that these criteria must be met after the 20th week of gestation, otherwise this would be classed as chronic hypertension.
3) Lines 125-126: subject-verb agreement error.
4) Lines 141-142: singular/plural error.
5) Line 166: "through activation NLRs." missing a word.
6) Line 271: "collecting" should be "collectively" perhaps.
7) Line 275: ASC-/- is incorrect nomenclature for a genetically modified mouse strain. It should be Asc-/- instead.
8) Line 296: Authors state "Li and colleagues demonstrated that the levels and distribution of MMPs..." but do not state how this protein changes in different conditions until subsequent sentences. This clause should be changed to "Li and colleagues quantified the levels and distribution of MMPs..."
9) Line 324: Labouring is spelled two different ways.
Round 2
Reviewer 2 Report
The authors have responded well in the body of the text by tempering the statement that the dysfunctional placenta is the cause of preeclampsia. They forgot however to do so in the first line of the abstract. I should recommend to change "The origins of preeclampsia (PE) lie within a functionally insufficient placenta" to "One of the important characteristics in the pathophysiology of preeclampsia is a dysfunction of the placenta."
Apart from this, I'm happy with the current version.
Author Response
we thank the reviewer pointing out this error. As suggested by this reviewer, we have made the changes in line 17 of the abstract submission.
We have requested Assistant Editor of the journal to allow us to edit the abstract section, which was submitted when submission process of was initiated.
Please see attached manuscript text, which incorporates the edited abstract.